# The Newly Synthetized Chalcone L1 Is Involved in the Cell Growth Inhibition, Induction of Apoptosis and Suppression of Epithelial-to-Mesenchymal Transition of HeLa Cells

**DOI:** 10.3390/molecules26051356

**Published:** 2021-03-03

**Authors:** Tomas Kuruc, Martin Kello, Klaudia Petrova, Zuzana Kudlickova, Peter Kubatka, Jan Mojzis

**Affiliations:** 1Department of Pharmacology, Faculty of Medicine, P. J. Safarik University, 04011 Košice, Slovakia; tomaskuruc@centrum.sk (T.K.); klaudia.petrova@student.upjs.sk (K.P.); 2Department of Chemistry, Biochemistry and Biophysics, University of Veterinary Medicine and Pharmacy, 04181 Košice, Slovakia; zuzana.kudlickova@uvlf.sk; 3Department of Medical Biology, Jessenius Faculty of Medicine, Comenius University in Bratislava, 03601 Martin, Slovakia; peter.kubatka@uniba.sk

**Keywords:** chalcones, antiproliferative, apoptosis, TGF-β, conditioned medium, epithelial-to-mesenchymal transition, tumour microenvironment, migration, HeLa cells

## Abstract

Over the past decades, natural products have emerged as promising agents with multiple biological activities. Many studies suggest the antioxidant, antiangiogenic, antiproliferative and anticancer effects of chalcones and their derivatives. Based on these findings, we decided to evaluate the effects of the newly synthetized chalcone L1 in a human cervical carcinoma cell (HeLa) model. Presented results were obtained by western blot and flow cytometric analyses, live cell imaging and antimigratory potential of L1 in HeLa cells was demonstrated by scratch assay. In the present study, we proved the role of L1 as an effective agent with antiproliferative activity supported by G2/M cell cycle arrest and apoptosis. Moreover, we proved that L1 is involved in modulating Transforming Growth Factor-β1 (TGF-β) signal transduction through Smad proteins and it also modulates other signalling pathways including Akt, JNK, p38 MAPK, and Erk1/2. The involvement of L1 in epithelial-to-mesenchymal transition was demonstrated by the regulation of N-cadherin, E-cadherin, and MMP-9 levels. Here, we also evaluated the effect of conditioned medium from BJ-5ta human foreskin fibroblasts in HeLa cell cultures with subsequent L1 treatment. Taken together, these data suggest the potential role of newly synthesized chalcone L1 as an anticancer-tumour microenvironment modulating agent.

## 1. Introduction

Cancer remains one of the leading causes of mortality globally. Although there is continual progress that has brought new methods in the treatment and prevention of cancer, a significant proportion of patients suffer from repeated relapses. This fact encourages increased efforts to understand the functioning of tumours and their microenvironment, as well as research into new treatments and the use of a wide range of anticancer drugs.

Tumours are highly complex structures that have been challenging the development of efficient anti-cancer therapies [1,2,3]. Heterogeneity within tumours acquires complexity during cancer progression, concurrently with the maturation of both: the cellular and non-cellular components. These components together create tumour niche known as tumour microenvironment (TME) [4,5]. The TME forms a complex system within tumours and is a major factor influencing the progression of cancer. TME plays a pivotal role in the regulation of cell growth, determines the metastatic potential, site of metastasis, and affects the outcome of subsequent treatment. The main representative of the non-cellular component is the extracellular matrix (ECM) followed by enzymes, growth factors, chemokines, and other signalling molecules that play a critical role in intercellular communication. In addition to malignant cells, stromal non-tumour cells (fibroblasts, mesenchymal stromal cells, endothelial cells, smooth muscle cells, pericytes, and adipocytes) and immune cells are another important part of the formation of TME [6]. Inhibition or stimulation of tumour growth is thus influenced by the different proportions of these components. Therefore, TME plays an important role in regulating tumour growth and metastasis.

While stromal cells themselves do not show malignant phenotypes, in the presence of TME, by interactions with each other and through direct or indirect interactions with cancer cells they acquire an abnormal phenotype and altered function. Their role in promoting tumour growth along with cancer cells is so important that they have become an attractive target for chemotherapeutic agents [7]. From the group of natural substances, chalcones appear to be suitable candidates for anti-cancer treatment.

Cervical cancer is ranked among top four malignancies for both incidence and mortality worldwide [8]. Still the leading cause of cervical cancer progression is HPV infection of epithelial cells and inefficient immune response within the cervical milieu [9]. Cervical cancer microenvironment can be described as final product of the effects based on high human papillomavirus (HPV) infection that in order instruct the surrounding stromal cells within the intercellular matrix. The TME is influenced by the paracrine activity of pre-cancerous and cancerous cells and its effect on the stromal cellular populations, mainly represented by fibroblasts/myofibroblasts and vice versa [10,11]. Thereby, it became clear that efficient cervical cancer therapy should take into account also the presence of cervical stromal cells. Natural substances that can inhibit tumour cells-stroma cells crosstalk by normalizing the TME may bring the new assets to the traditional tumour therapy.

Chalcones have been reported as precursors of flavonoids and isoflavonoids, belonging to the group of polyphenols and being biosynthesized in numerous plant species. Chemically, chalcones are described as (*E*)-1,3-diphenyl-2-propen-1-ones consisting of two aromatic rings joined by a three-carbon α,β-unsaturated carbonyl system [12]. The biological effects of chalcones and their derivatives have been described in numerous studies showing their anti-inflammatory [13,14], antioxidant [15,16], anti-fungal and anti-microbial [17], anti-angiogenic [18,19], and immunomodulatory effects [20]. Moreover, chalcones exhibit promising anti-neoplastic/anti-proliferative activities [21,22,23]. The cytotoxic effects of chalcones are exerted through various mechanisms including disruption of the cell cycle, inhibition of tubulin polymerization, induction of apoptosis [21,24,25,26], interaction with multiple kinases [27], multidrug-resistance proteins [28,29] and various signalling pathways involved in cell survival and death [21,30,31,32]. These facts are showing the potential of chalcones in targeting TME.

One of the important ways in which are chalcones able to influence TME is disrupting the tumour vasculature (as described in our previous review [18]) and/or through changes in components of ECM. The several evidences exist, that chalcones are involved in regulation of matrix metalloproteinases 2 and 9 (MMP-2 and MMP-9) activity [33,34]. Moreover, chalcones modulate expression of epithelial markers, such as E-cadherin and mesenchymal marker genes such as N-cadherin and vimentin [35,36,37]. As we already know, low levels of E-cadherin are associated with increased metastatic potential [38]. On the other hand, higher levels of N-cadherin and vimentin contribute to the process of metastasis and invasiveness [39,40]. Described chalcone effects, in the end, lead to the weakening of invasiveness and metastatic potential.

Under the influence of published data, the aim of this study was to evaluate the potential anti-cancer effect of newly synthesized chalcone L1. We focused on the antiproliferative effect, induction of apoptosis, and interfering with signalling pathways along with proteins involved in migration and metastasis in cervical cancer model.

## 2. Results

### 2.1. Cell Viability Assay

To validate changes in cell viability (Figure 1A), HeLa cells were treated with L1 (Figure 1B) and MTS assay was performed. The presented results clearly showed that the tested compound significantly decreased the viability of seeded HeLa cells in a concentration-dependent manner. The obtained value of IC_50_ is represented by a 10 µM concentration of L1. Therefore, this concentration was used in subsequent analyses along with the “non-toxic” concentration IC_10_ 1 µM. Moreover, concentrations higher than the value of IC_50_ did not show any further significant decrease in cell viability.

### 2.2. Cell Cycle Analysis

To evaluate the cell cycle distribution of HeLa cells treated with L1, flow cytometry analyses were performed. As seen in Figure 2 and Table 1, treatment with L1 after 24 h significantly (*p* < 0.05) induced cell cycle arrest in the G2/M phase, and this effect was followed by a decreasing trend at 48 h and 72 h compared to control. Moreover, we noticed a significant increase in sub-G0 population of cells with the highest peak at 72 h after L1 treatment. This population is typically considered as apoptotic cells. Results suggest that treatment with L1 contributes to cell cycle arrest at the G2/M phase and induces apoptosis in a time-dependent manner.

### 2.3. Analysis of Apoptosis Occurence

To determine pro-apoptotic potential of L1, we performed flow cytometric analyses of several proteins involved in this process. As shown in Figure 3, we evaluated the activation of Caspase-3 (A) and cleavage of PARP (Poly (ADP-ribose) polymerase) (B). Results showed, that L1 alone or in combination with TGF-β was capable of significantly increase Caspase-3 activation and cleavage of PARP compared to untreated control groups as well as TGF-β. This effect persisted after 48 h and 72 h suggesting the pro-apoptotic effect of L1. Analyses of Annexin V/PI double staining displayed in Table 2 further confirm this effect, showing significant increase of early apoptotic cell populations and concomitant decrease in living cells after 48 h and 72 h of treatment. TGF-β co-treatment with L1 showed that pro-apoptotic effects of L1 were minimally or no affected by TGF-β presence.

### 2.4. Analysis of Proliferation Activity of HeLa Cells

The antiproliferative effect of L1 was determined by flow cytometric analyses after cell staining with CellTrace^TM^ Yellow. The untreated control cell division is presented in Figure 4A. Results in the Figure 4B below showed, that L1 significantly inhibited cell proliferation compared to the untreated control group at 72 h. The proliferation rate after short time TGF-β treatment, as seen in the Figure 4C, did not significantly surpass or suppress the proliferation compared to the control groups. This effect has been observed throughout all analysed times and was comparable to control groups displayed in Figure 4A. Moreover, individual treatments as well as combination treatment at 72 h were merged and compared in Figure 4D. In the case of combination treatment (TGF-β and L1), the tested compound markedly suppressed the positive proliferative effect of TGF-β on HeLa cells. Similarly, this effect was most visible after 72 h.

### 2.5. Effect of L1 and TGF-β Treatment on Selected Proteins

#### 2.5.1. Changes in Expression and Phosphorylation of Smad2/3 Proteins as the Main Signal Transducers for TGF-β Receptors

It is well known that family of Smad proteins is closely related to TGF-β signalling, particularly Smad2/3, that promote signal from activated receptors further to the cell. Results displayed in Figure 5A,B indicate changes in both, total and phosphorylated form. A single treatment with TGF-β met our expectations and resulted in a significant increase of expression and phosphorylation compared to control groups, except of total Smad2/3 at 48 h. In the case of a single L1 treatment or its combination with TGF-β, we can observe a significant decreasing trend related mainly to the phosphorylation of Smad2 at 48 h and 72 h. Similar effect was detected in relative expression of total Smad2/3 at 72 h, however there was not any significant difference between combination treatment and single L1 treatment. It is noticeable that L1 + TGF-β treatment was not so efficient in suppressing Smad2 phosphorylation as L1 alone (*p* < 0.001) at both, 48 h and 72 h. Moreover, a single L1 treatment significantly decreased both, total Smad2/3 and phosphorylation of Smad2 compared to control groups at each time.

#### 2.5.2. Changes in Expression and Phosphorylation of MAPK Proteins Along with Akt Activation

It is well known that proteins involved in the MAPK signalling pathway play a crucial role in transduction extracellular signals to cellular responses including proliferation, differentiation, development, transformation, cell survival, and apoptosis. There is an intimate crosstalk between MAPK and PI3K/Akt signalling pathway, which is also involved in the abovementioned responses.

We decided to investigate the effect of L1 on specific proteins of MAPK and PI3K/Akt signalling pathways such as Erk1/2, p38 MAPK, JNKas well as Akt in HeLa cells. Results shown in Figure 6A–C indicate that the treatment with TGF-β significantly increased both, total levels of proteins and their phosphorylated forms in most cases. However, this effect was visible mainly after 72 h of incubation compared to control groups, which indicates the time-dependent effect of TGF-β treatment. On the other hand, results show that treatment with L1 alone or in combination with TGF-β significantly suppressed the effect of single TGF-β treatment mainly on phosphorylation of individual proteins. This effect was visible throughout the analyses at both 48 h and 72 h with exception of phospho p38 MAPK, JNK and total JNK at 48 h. Moreover, L1 alone or in combination with TGF-β significantly decreased expression and phosphorylation of individual proteins in most cases (Erk1/2, Akt) compared to control groups.

On the other hand, the only significant exception in abovementioned results (L1 and L1 + TGF-β treatment) is increased relative expression and phosphorylation of p38 MAPK and JNK. Results indicate that treatment with L1 and L1 + TGF-β significantly surpassed control groups in time-dependent manner. Finally, the single treatment with L1 possesses a stronger inhibitory effect compared to control groups, TGF-β treatment, and combination treatment (phospho Erk1/2, Akt) with a few time-dependent exceptions in case of p38 MAPK, JNK or total Erk1/2 at 72 h in HeLa cells model.

#### 2.5.3. Changes in Expression of Selected Proteins Involved in Migration and Invasiveness

It is widely known, that cell surface proteins as N- and E-cadherins play crucial role in cell-cell adhesivity, migration, and invasiveness, which increase metastatic potential of cancers. Specifically, the down-regulation of E-cadherin followed by up-regulation of N-Cadherin contributes to the process of epithelial-to-mesenchymal transition, which is a hallmark of metastasis. Along with these cell surface proteins, members of the matrix metalloproteinases family, particularly MMP-9 greatly contributes to the metastatic potential. 

Results in Figure 7A,B,D above suggest, that TGF-β treatment significantly increased expression of N-cadherin and MMP-9 compared to control groups of HeLa cells. This effect was achieved in time dependent manner with highest expression rate at 72 h. However, levels of both proteins were significantly reduced after combination (L1 + TGF-β) and single treatment with L1 compared to control group as well as to TGF-β group.

In case of E-cadherin expression, we expected its TGF-β-induced down-regulation. Results proved our expectation as seen in Figure 7C, where TGF-β significantly supressed E-cadherin expression compared to control group in HeLa cells. The opposite effect was detected in case of combination as well as single L1 treatment. L1 possessed significant antagonistic effect in TGF-β induced E-cadherin suppression. The most prominent effect related with up-regulation of E-cadherin expression was detected after a single L1 treatment with the highest level at 72 h.

### 2.6. Analyses of HeLa Treatment with Conditioned Medium and L1

In solid tumours, stromal cells are in immediate contact with tumour parenchyma. These non-malignant cells secrete various growth factors, metabolites, and extracellular matrix proteins to the tumour microenvironment. Thus, we tested whether conditioned medium (CM) from fibroblast (BJ-5ta) alone or in combination with L1 possessed any significant effect on expression and phosphorylation of selected groups of proteins (Figure 8, Figure 9 and Figure 10) in HeLa cells.

#### 2.6.1. Effect of Conditioned Medium on Expression and Phosphorylation of Smad2/3 Proteins as the Main Signal Transducers for TGF-β Receptors

The analyses of single CM treatment suggest that it is capable of significantly increase expression of total Smad2/3 proteins as well as phosphorylation of Smad2 compared to control groups (Figure 8A,B) in HeLa cells. This effect persisted at 48 h and 72 h. However, the treatment was more effective at 48 h, mainly in case of total Smad2/3. Combination treatment (CM + L1) resulted in significant decrease of either expression or phosphorylation compared to treatment with CM, but results kept close to control groups. The most significant effect in the decrease of both, total Smad2/3 and phosphorylation of Smad2 was achieved in a single L1 treatment as detected in the results above. It is clear that L1 attenuated effects of CM treatment in HeLa cells.

#### 2.6.2. Effect of Conditioned Medium on Phosphorylation of MAPK Related Proteins along with Akt Activation

Throughout all analyses except of phosphorylation of p38 MAPK, CM treatment on HeLa cells achieved a significant increase in phosphorylation of individual proteins compared to control groups at both, 48 h and 72 h (Figure 9A–C). In case of p38 MAPK, treatment with CM significantly inhibited its phosphorylation (Figure 9D).

The L1 single treatment in HeLa cells showed significantly decreased activation of Akt and Erk1/2 at 48 h and 72 h compared to control groups, while activation of JNK and p38 MAPK reached the opposite effect. On the other hand, combination treatment had a different effect on phosphorylation of Akt and Erk1/2 (A, B) vs. phosphorylation of JNK and p38 MAPK (C, D). Phosphorylated forms of Akt and Erk1/2 were significantly decreased in time-dependent manner under effect of combination treatment compared to groups treated with CM. The opposite increased effect was achieved in case of JNK and p38 MAPK levels in combination or L1 treatment compared to CM group. Results showed significant differences between combination treatment *versus* single L1 treatment among individual proteins. But it is clear that chalcone L1 in combination with CM significantly modulated effect of CM.

#### 2.6.3. Effect of Conditioned Medium on the Expression of Proteins Involved in Migration and Invasiveness

As seen in Figure 10, effect of CM influenced levels of N-cadherin (A), E-cadherin (B), and MMP-9 (C) in HeLa cells. CM significantly contributed to up-regulation of N-cadherin and MMP-9 compared to control groups at 48 h and 72 h with non-significant time dependency. On the other hand, E-cadherin was significantly time-dependently down-regulated 72 h after CM treatment.

In CM experiments, the effect of L1 on selected proteins showed the same described effects as we observed in TGF-β experimental part (Results 2.4.3). The L1 significantly reduced N-cadherin and MMP-9 expression and concurrently increased E-Cadherin expression at 72 h after treatment. These parallel experiments also showed consistence of L1 effects in HeLa cells.

Furthermore, we expected that chalcone L1 in combination treatment will down-regulate the levels of N-cadherin and MMP-9 compared to single CM treatment. Results proved our expectation and showed significant differences not only between CM groups, but even compared to control groups mainly at 72 h. Moreover, this down-regulation possessed a time-dependent manner with highest inhibition rate at 72 h. In case of E-Cadherin the opposite increased effect was achieved, where combination treatment still caused its significant up-regulation, showing a time-dependent manner with highest expression rate at 72 h. Comparing L1 and combination treatment, regarding the effect on E-cadherin, there is a significant difference, where L1 alone had a greater potential to up-regulate its levels. In addition, the time-dependent manner was observed with the highest difference at 72 h.

### 2.7. Scratch Assay

To find out whether L1 has the potential to suppress cell migratory properties, a scratch/wound healing assay was performed on HeLa cells. As seen in the figure below, cells in the control groups gradually repopulated the wounded area in a time-dependent manner (Figure 11A). We noticed a similar effect in groups treated with a non-toxic concentration of L1 (1 µM) except for 24 h, where the ability of cells to repopulate the wounded area was temporarily attenuated compared to the control group. However, it is noticeable that cells started to repopulate this area more efficiently at 48 h or 72 h. Nevertheless, the difference remained significant compared to the control group. In contrast to previous groups, 10 µM concentration of L1 significantly reduced wound healing properties of seeded cells. This effect was preserved within the whole incubation process with the highest difference at 72 h compared to the control group as well as with the group treated with non-toxic concentration. On the other hand, groups treated with TGF-β did not show any significant increase in repopulating of the wounded area and the captured photos are comparable to control groups. This result is in accordance with proliferation analyses that also showed no visible acceleration of proliferation of HeLa cells. The combination treatment (TGF-β + L1 1 µM or L1 10 µM) followed similar trend as in the case of single L1 treatment with both concentrations, showing that L1 (10 µM) is capable of significantly reducing the repopulation of wounded area even in the presence of TGF-β. However, we suggest that in this concentration migration may be affected by antiproliferative/cytotoxic effect of chalcone L1.

### 2.8. Co-Cultures Staining

Under normal circumstances, solid tumours are composed of two main parts, including parenchyma and stroma. Non-malignant cells along with extracellular matrix compose the stroma, whereas cancer cells belong to the parenchyma. To partially simulate interactions in solid tumours, HeLa cells and non-malignant BJ-5ta cells were co-cultivated. These co-cultures were stained after 72 h to visualize levels of N-Cadherin and Smad2/3 with or without L1 (10 µM) treatment. As seen in Figure 12A,B, treatment with L1 significantly decreased levels of both, N-cadherin and Smad2/3 compared to control groups.

## 3. Discussion

The complexity of cancer biology was firstly highlighted by six main hallmarks: insensitivity to anti-growth signals, self-sufficiency in growth signals, evading apoptosis, limitless potential to replicate, continuous angiogenesis, and invading of tissues with subsequent metastasis [41]. Later, two more hallmarks have been added: energy metabolism reprogramming and evading the immune response [42]. However, solid tumours are considered as a highly heterogeneous and complex organs, where not only cancer cells play a crucial role. Non-malignant cells of the tumour mass possess a dynamic function and often support tumour growth during all stages of carcinogenesis [43]. Thus, we have to think of tumours with not only focus on the cancer cells but also with the consideration of their complex tumour microenvironment. In content of cervical cancer, TME affecting carcinogenesis (after HPV infection) by several ways including locally increased levels of cytokines, particularly TGF-β1 and IL-10 [44]. It has also been reported that cytokine/growth factor TGF-β1 promotes chromosomal instability in cervical epithelial cells infected by HPV [45] and mediated processes related to cervical cancer invasion [46] or pelvic lymph node metastasis in early-stage [47]. The TME represented by pericytes, adipocytes, endothelial cells, fibroblasts and myofibroblasts, bone-marrow-derived mesenchymal stem cells (BM-MSCs), mesothelial cells and leucocytes is a massive field that significantly affects cell crosstalk, cell-to-cell adhesion and neo-angiogenesis and de facto tumour development and metastasis.

Therefore, the potential drug with anti-cancer properties have to interfere with one or more of the above-mentioned traits. Chalcones, as natural compounds belonging to the group of polyphenols, with their biological effects already mentioned in our paper above, appear to be promising agents for cancer intervention.

One of the typical features of tumour cells is their uncontrolled division [48,49]. Therefore, substances interfering with the cell proliferation are promising tool in the suppression of tumour growth. The newly synthesized chalcone L1 is also such a substance. Our results point to the fact, that L1 was effective as antiproliferative agent alone or even in combination with TGF-β. However, as our results showed, we did not notice any significant changes in proliferation after single TGF-β treatment. On the other hand, TGF-β is capable of either suppressing or stimulating the proliferation depending on the type of cell lines, working concentrations or duration of the treatment [50]. Anyway, the highest antiproliferative effect of chalcone L1 was detected after 72 h of treatment. Many other studies also suggest the antiproliferative effect of chalcones through various mechanisms in multiple cancer cells lines [21,22,24,51,52].

The antiproliferative effect itself is directly linked with changes in cell cycle and overall survival of cells. More specifically, the suppression of proliferation is caused by cell cycle arrest and induction of apoptosis [53,54]. Cell cycle progression through all four phases (G1, S, G2 and M) is critical for cell growth. To ensure proper progression of cell cycle, two important control mechanisms (G1/S and G2/M checkpoints) are involved in controlling of unlimited growth. Cancer cells usually develop a defective G1/S checkpoint by down-regulation of tumour suppressors as p53 or Rb, whereas G2/M checkpoint is often intact, thus crucial in survival of cancer cells [55]. To determine proliferation activity and cell cycle distribution, we used flow cytometry analyses. As our results showed, L1 was able to inhibit the proliferation by inducing G2/M phase cell cycle arrest already after 24 h of incubation. The G2/M arrest is commonly associated with the induction of apoptosis [21,24,56]. This fact coincided with our subsequent findings, where L1 induced increase of sub-G0 population after G2/M phase arrest. Accumulation of cells in sub-G0 population subset is often considered as marker of apoptosis [57,58], which supports our findings. According to our results, induction of apoptosis was confirmed by significant increase in caspase-3 activation after L1 treatment at 48 h and 72 h. Activation of caspase-3 was further confirmed by significant increase in cleavage of PARP. Moreover, Annexin V/PI double staining showed the decrease of living cells and increase of early apoptotic cells mainly at 72 h after treatment in Hela cells population. These results were consistent with those obtained by Takac et al. [21,22] and Kello et al. [24], even though Takac et al. detected mainly late apoptotic cells after Annexin V/PI staining following the chalcone 1C treatment. Moreover, increase in sub-G0 population, externalisation of PS, caspase-3 activation and PARP cleavage followed the time-dependent manner with highest rate after 72 h of incubation. The exact underlying mechanisms in L1 induced cell cycle arrest and apoptosis are not clear since it was not the priority of this study. We assume, that L1 interfered with microtubule formation. This phenomenon concerning newly synthesized chalcones has been already demonstrated in several studies performed by us and other researchers [21,59,60].

The TGF-β is a multifunctional cytokine involved in regulation of fundamental cellular processes such as differentiation, proliferation, morphogenesis, regeneration, and stem-cell maintenance [61]. The main signal transducers for TGF-β receptors are Smad proteins, which represent Smad-dependent TGF-β signalling [62]. We decided to test L1, whether it is capable of suppressing Smad-mediated TGF-β signalling. As our results showed, L1 effectively suppressed activation of Smad2 protein together with decreasing total levels of Smad2/3. Recently Jeong et al. achieved the same result with their tested chalcone in suppression of Smad2 phosphorylation [63]. In another study, Hseu et al. reported a similar effect of chalcone flavokawain A on phosphorylation and transcriptional activity of Smad3 [64]. Furthermore, we demonstrated the reduction in total Smad2/3 levels after L1 treatment even by immunofluorescent staining of co-cultures.

Another way through which TGF-β mediates cellular responses is Smad-independent pathway. The main signal transducers of this pathway comprise the mitogen-activated protein kinase (MAPK) pathway consisting of p38 MAPK, c-Jun amino terminal kinase (JNK), and extracellular signal-regulated kinases (Erk1/2) along with the phosphatidylinositol-3 kinase (PI3K)/Akt pathway [65]. It is widely known that MAPKs play a crucial role in regulation of various cellular processes such as cell survival, proliferation, and apoptosis [66]. Activation of Erk1/2 has been linked with cell survival and proliferation [67,68], however several studies demonstrated its activation during process of apoptosis induced by the treatment with natural compounds [69,70]. Similarly, the role of activated p38 MAPK is commonly associated with apoptosis induced by polyphenols [71], which also includes chalcones. Finally, the Akt as component of PI3K/Akt signalling pathway is involved in regulation of apoptosis, cell survival, and progression of cell cycle [72]. Our aim was to study the effect of L1 on individual proteins involved in these signalling pathways. Results show that L1 effectively decreased levels of phosphorylated Erk1/2 even after TGF-β stimulation. On the other hand apoptotic phosphorylation of p38 MAPK was rapidly increased after L1 treatment at 72 h. Moreover, L1 significantly suppressed the positive effect of TGF-β on p38 MAPK phosphorylation, where TGF- β act as epithelial-to-mesenchymal transition (EMT) modulator [73,74]. As the several studies have described, the activation or deactivation of Erk1/2 is associated with apoptosis depending on experimental conditions and type of cell line. As we experimentally confirmed, Erk1/2 activation was inhibited by L1 treatment. Similar effect demonstrated Yuan et al., where their tested compound significantly suppressed activation of Erk1/2, and increased activation of p38 MAPK. These findings were associated with induction of apoptosis [75]. Looking back to our results, we detected G2/M arrest and subsequent apoptosis after L1 treatment as well, which correlates with their findings. In case of Akt, L1 proved to be an inductor of its deactivation. L1 not only reduced the phosphorylation compared to control, but significantly decreased the activation compared to positive control group treated with TGF-β.

The apoptosis and tumour cell survival is associated also with JNK activation [76]. The results showed time-dependent increase of JNK phosphorylation after L1 treatment. Slight increase was marked also after TGF-β but L1 co-treatment was able after 72 h suppressed the TGF-β effect. This further confirm the involvement of L1 in apoptosis and cell cycle arrest, which was also demonstrated by other studies [77,78]. Altogether, chalcone L1 was able to promote G2/M arrest, inhibit proliferation, and induce apoptosis by modulating the components of MAPK and PI3K/Akt signalling pathways.

The TGF-β possess different effects at various stages of cancer development. In early stages it acts as a potent tumour suppressor and growth inhibitor of epithelial cells. On the other hand, in advanced stages it is involved in tumour growth and progression by inducing EMT, leading to tumour invasion and metastasis [61,79]. To this date, many EMT-related markers have been described [80,81]. The process of EMT evolve in several steps including the loss of epithelial marker E-cadherin, up-regulation of mesenchymal markers as N-cadherin and vimentin, executing to increased migratory potential and invasiveness [82]. Along with these changes, the increased cell motility is also supported by the secretion of MMP-2 and MMP-9 [83]. According to these facts, we decided to test the effect of L1 in suppression of EMT process. Surprisingly, L1 was effective in down-regulation of EMT related proteins, namely N-cadherin and MMP-9. Similarly, as discussed above, L1 also significantly suppressed the effect of TGF-β, which promoted the up-regulation of these markers. Moreover, the expression of epithelial marker represented by E-cadherin was recovered after L1 treatment as well. Here, TGF-β stimulation resulted in decrease of E-cadherin levels. These results suggest that our tested compound has a potential to suppress EMT. Many other studies confirm similar effect of chalcones in the process of EMT [33,36,63,84,85].

In addition, the involvement of L1 in suppression of TGF-β-induced EMT is supported by above-mentioned results related with Smad-dependent as well as Smad-independent pathways. As the role of TGF-β in EMT was clearly recognized, disrupting TGF-β signalization further suppress this process. We demonstrated, that L1 effectively prevented phosporylation of Smad-2 and Erk1/2 as the main signal transducers in Smad-dependent or independent pathways. Many other authors already described that interfering with the phosphorylation of Smad2 or Smad3 is crucial in blocking TGF-β/Smad signalling, thus leading to EMT suppression [85,86,87]. Zavadil et al. together with Xie et al. reported Erk1/2 activation as another essential step in TGF-β induced EMT, and it is also a requirement for disassembling of adherens junctions and cell motility [88,89]. These findings correlate with our results discussed above. In addition, the activation of Akt as a component of PI3K/Akt pathway was significantly suppressed as well. Xue et al. have already described the contribution of Akt activation in the process of EMT [90]. In contrast, p38 MAPK activation possess an EMT promoting role [91,92]. Our results showed increased activation status of p38 MAPK in all treated groups. However, L1 proved to be effective in decreasing TGF-β-induced p38 MAPK activation. Elevated levels of phosphorylation in L1 single treatment, may be caused by induction of apoptosis and increase in relative expression of total p38 MAPK, as seen in the results section. Finally, we further supported the effect of L1 in potential suppression of EMT by down-regulation of N-Cadherin after co-cultures staining.

Conditioned media (CM) are defined as a collection of various proteins containing signal peptides. Generally, it is considered as a secretome encompassing cell surface proteins and intracellular proteins secreted through multiple pathways. The proteins found in conditioned media include growth factors, cytokines, enzymes, hormones and other soluble mediators involved in the processes of differentiation, cell growth, proliferation, invasion and angiogenesis [93]. Stromal fibroblasts represent the essential component of the tumour microenvironment in promoting of growth and invasiveness of cancer cells through various mechanisms [94,95,96]. Therefore, we decided to use CM obtained from fibroblasts for HeLa cells cultivation. As seen in the results section, effect of CM on Smad2 activation was closely related to this of TGF-β. This was probably due to elevated levels of various growth factors and cytokines contained in CM. However, similarly as in the case of TGF-β, L1 was able to reduce phosphorylation status of Smad2, thus impairing the downstream Smad signalling. Fullar et al. reported that fibroblasts effectively produced TGF-β, which in turn promoted cell proliferation [11]. This only further confirms the antiproliferative effect of L1 proven by flow cytometry.

As we already discussed, TGF-β signalling is also mediated through Smad-independent pathway. Therefore, we evaluated the effect of CM on members of MAPK and PI3K/Akt signalling pathways as well. We found that CM increased the phosphorylation of Erk1/2, JNK, and Akt, except for p38 MAPK. While L1 suppressed the activation of Erk1/2 and Akt, it promoted activation of JNK and p38 MAPK. The effect of CM from stromal cells on these proteins has been elucidated as well. Ko et al. reported that CM increased the phosphorylation of Akt, and pharmacological intervention inhibited CM-induced Akt activation, thus blocking the proliferation of cancer cells [97]. The similar effect was achieved by Gao et al., where CM from stromal fibroblasts increased the activation of Erk1/2 [98]. Moreover, the synchronous activation of both, Erk1/2 and PI3K/Akt was found to be associated with increased extracellular matrix and collagen production [99]. Both are important components of tumour microenvironment. We have already discussed above that Erk1/2 is a crucial signal transducer for EMT, cell survival and proliferation. Blocking Erk1/2 by L1 thus contributes to suppression of tumour growth and metastasis. It has been proposed that JNK activation leads to apoptosis induced by different stressors [76]. Although CM slightly increased the activation of JNK, L1 surpassed this effect indicating apoptosis. Similar effect was achieved in p38 MAPK activation, the role of which we already discussed. The role of fibroblasts in cell proliferation of cancer cells was also evaluated by Koh et al. and Steer et al. [100,101]. Their results suggest the positive/negative antiproliferative effect of fibroblasts on cancer cells depending on fibroblasts type, cancer cells types and their combinations.

The effect of CM has also been demonstrated on proteins involved in migration and invasiveness. We already discussed the involvement of E-cadherin, N-cadherin and MMP-9 in these processes. Our results point to the fact that CM possessed a similar effect as TGF-β. It effectively increased levels of N-cadherin and MMP-9 but decreased the level of E-cadherin. On the other hand, L1 has already proved the opposite effect. These results suggest that CM substances composition, similarly to TGF-β, could be possibly involved in the process of EMT as well. Many other studies reported that fibroblast secretome significantly contributed to migration and EMT-related phenotype changes in cancer cells. For instance, using a co-culture system of fibroblasts and cancer cells, changes in the gene expression involved in EMT have been proposed [102] along with increase of cell growth [103] and MMP-9 [104]. The study performed by Limoge et al. revealed that fibroblasts induced growth of breast carcinoma by stimulating the tumour vasculature through the MMP-9-dependent mechanism [105]. Promoting tumour vasculature is another important role of MMP-9. In addition, Gao et al. observed up-regulation of N-cadherin and down-regulation of E-cadherin in co-cultures of fibroblasts and cancer cells [98]. These studies along with our results support the role of fibroblasts in promoting cancer cell growth, proliferation and EMT. However, L1 was able to suppress effect of CM. The fact that L1 potentially suppress cancer cell migration was further proved also by scratch assay, where the concentration-dependent effect was observed. This finding correlate with our results discussed above.

## 4. Materials and Methods

### 4.1. Cell Culture

The Cell line HeLa (human cervical adenocarcinoma) and BJ-5ta (human dermal fibroblasts) were obtained from ATCC (Manassas, VA, USA). HeLa cells were cultured in growth medium RPMI 1640 (Biosera, Kansas City, MO, USA) supplemented with a 10% fetal bovine serum (FBS) (Invitrogen, Carlsbad, CA, USA) and 1× HyClone™ Antibiotic/Antimycotic Solution (GE Healthcare, Piscataway, NJ, USA). BJ-5ta cells were cultured in DMEM-M199 4:1 medium mixture and supplemented with 10 % FBS and hygromycin B (0.01 mg/mL). Cells were maintained in standard conditions with an atmosphere containing 5% CO_2_ at 37 °C. Prior to each experiment, cell viability was greater than 95%.

### 4.2. Tested Compound

Chalcone L1 ((2*E*)-1-(1-methoxy-1*H*-indol-3-yl)-3-(2,4,6-trimethoxyphenyl)-prop-2-en-1-one, Figure 1B) was synthesized and characterized by Kudlickova et al. (Department of Chemistry, Biochemistry and Biophysics, University of Veterinary Medicine and Pharmacy, Košice, Slovakia [106] (where tested chalcone L1 is identified as compound 12b). The tested compound was dissolved in dimethyl sulfoxide (DMSO) with the final concentration of DMSO 0.02% in the growth medium in each experiment. The final concentration of DMSO exhibited no cytotoxicity on cultured cells.

### 4.3. Cell Viability Assay

The antiproliferative effect of L1 was determined by MTS colorimetric assay ((3-(4,5-dimethylthiazol-2-yl)-5-(3-carboxymethoxyphenyl)-2-(4-sulfophenyl)-2H-tetrazolium)). Tested HeLa cells (5 × 10^3^/well) were seeded in 96-well plates. After 24 h, L1 final concentrations (1–30 µM) were added and incubation proceeded for the next 72 h. In the next step, 10 μL of MTS (5 mg/mL, Sigma Aldrich, St. Louis, MO, USA) was added to each well containing cells and incubated for another 2 h at 37 °C. After 2 h, the metabolic activity of cells was evaluated by measuring the absorbance at wavelength 490 nm using the automated Cytation™ 3 Cell Imaging Multi-Mode Reader (Biotek, Winooski, VT, USA). The absorbance of wells containing control groups was taken as 100% and the results were expressed as fold of the control.

### 4.4. Conditioned Medium

Human Dermal Fibroblasts (BJ-5ta) (ATCC) were cultivated until confluence, rinsed 1x with sterile PBS, and cultured in starving medium—DMEM/M199 mixture without FBS or antibiotic/antimycotic solution. After 24 h, conditioned medium (CM) from these cells was harvested, centrifuged (1200 rpm/5 min) and the supernatant was filtered (syringe filters 0.22 µm) to remove cells and debris. Filtered CM was then aliquoted and stored at −80 °C for experiments.

### 4.5. Flow Cytometric Analyses

According to the experimental scheme, two main groups of conditions were created. In the first group, HeLa cells were cultivated only in a complete medium with 10% FBS and treated with L1 (10 µM) and TGF-β (10 ng/mL) or their mutual combinations. The second group included conditions in which cells were cultivated in addition with CM and treated with L1 (10 µM) or their combinations. HeLa cells were seeded in Petri dishes with complete growth medium and cultivated for 24 h. After 24 h, cells were treated with L1 and depending on the experimental scheme, TGF-β or CM was added. Adherent and floating cells were harvested, washed in PBS, divided for subsequent analysis, stained with conjugated antibody or with primary antibody (Table 3) and incubated for 20 min (room temperature, in the dark), followed by staining with secondary conjugated antibody if needed (20 min). After incubation, fluorescence was detected using BD FACSCalibur flow cytometer (BD Biosciences, San Jose, CA, USA). A minimum of 1 × 10^4^ events were analyzed per sample.

### 4.6. Cell Cycle Analysis

For this purpose, adherent and floating HeLa cells were harvested three different times (24, 48 and 72 h) after treatment (L1 10 µM and TGF-β 10 ng/mL or CM), washed in cold PBS, fixed in cold ethanol (70%) and stored at −20 °C until analysis. Prior to each analysis, cells were washed in PBS followed by resuspending in staining solution (0.5 mg/mL ribonuclease A, 0.2% final concentration Triton X-100, propidium iodide 0.025 mg/mL in 500 µL PBS) (all from Sigma Aldrich, St. Louis, MO, USA) and incubation for 30 min at room temperature in the dark. For analysis of stained cells, a BD FACSCalibur flow cytometer (BD Biosciences, San Jose, CA, USA) was used. A minimum of 1 × 10^4^ events were analyzed per analysis.

### 4.7. Analysis of Apoptosis

To perform these analyses, HeLa cells seeded in Petri dishes were treated with L1 (10 µM) and TGF-β (10 ng/mL) for 24, 48 and 72 h. After the treatment, adherent and floating cells were harvested, centrifuged and pellet was resuspended in PBS. Resuspended cells were subsequently divided for individual staining (Caspase-3, PARP and Annexin V/PI double staining). After 20 min of incubation with mentioned primary antibodies at room temperature in the dark, 1 µL of propidium iodide (PI) (0.025 mg/mL) was added to the samples stained with Annexin V antibody. For analysis of stained cells, a BD FACSCalibur flow cytometer (BD Biosciences) was used. A minimum of 1 × 10^4^ events were analyzed per analysis.

### 4.8. Cell Proliferation Analysis

To analyse the proliferation activity of HeLa cells, 5 mM stock of CellTrace^TM^ Yellow Cell Proliferation Kit for flow cytometry was used (Thermo Scientific). Cells were harvested, centrifuged and pellet (1 × 10^6^ cells) was resuspended in 1 mL staining solution (2 µL CellTrace ^TM^ Yellow in 1 mL PBS) to reach 10 µM working concentration. Cells were incubated for 20 min at 37 °C in the dark. After incubation, cells were resuspended in a complete culture medium and incubated again for 5 min at 37 °C. Following second incubation, cells were centrifuged to remove supernatant, and the pellet was resuspended in fresh, pre-warmed complete culture medium and seeded in Petri dishes. Afterwards, seeded cells were divided into specific conditions for subsequent analysis. Cells were treated with L1 (10 µM) and analyzed in three different times (24 h, 48 h, and 72 h). For analysis of stained cells, a BD FACSCalibur flow cytometer (BD Biosciences) was used. A minimum of 1 × 10^4^ events were analyzed per analysis.

### 4.9. Western Blot Analyses

HeLa cells were treated with tested compound L1 (1 µM and 10 µM) and TGF-β (10 ng/mL) or their combinations for 24 h, 48 h, and 72 h. Laemli lysis buffer containing glycerol, 1 mol/L Tris/HCl (pH 6.8), 20% sodium dodecyl sulfate (SDS), and deionized H_2_0 in the presence of PIC and PhIC (protease and phosphatase inhibitor cocktail, Sigma Aldrich) was used for preparing protein lysates followed by a process of sonication. The protein concentration was determined using the Pierce^®^ BCA Protein Assay Kit (Thermo Scientific) and measured using an automated Cytation™ 3 Cell Imaging Multi-Mode Reader (Biotek) at wavelength 570 nm. For protein separation, SDS-PAA gel (10%) was used at 100 V for 2 h. Proteins were then transferred to PVDF Blotting membrane using iBlot™ 2 Dry Blotting System (Invitrogen, Carlsbad, CA, USA). The membrane with transferred proteins was then blocked (1 h, room temperature) in 4% dry non-fat milk or 4% BSA with TBS-Tween (pH 7.4), depending on the primary antibody used, to minimize non-specific binding. Blocking was then followed by incubation with primary antibodies over-night at 4 °C. Immunoblotting was carried out with the antibodies mentioned below (Table 4). On the following day, membranes were washed in TBS-Tween 3 times for 5 min and then incubated with corresponding horseradish peroxidase-conjugated secondary antibodies for 1 h at room temperature. After incubation, washing with TBS-Tween (3 × 5 min) was again performed on membranes. Protein expression was detected using MF-ChemiBIS 2.0 Imaging System (DNR Bio-Imaging Systems, Jerusalem, Israel) with chemiluminescent ECL substrate (Thermo Scientific).

### 4.10. Scratch Assay/Wound Healing Assay

Scratch assay was used for evaluating of migratory properties of tested cells under specific conditions. HeLa cells (3.5 × 10^5^) were seeded into 6-well plates and cultured in a growth medium until they reach confluence. A linear scratch through cell monolayer was made in individual wells using an SPL Scar™ scratcher (SPL Life Science, Pocheon, Korea). Each well was then gently washed 2 times with 3 mL of PBS and treated with L1 (1 µM and 10 µM) in the growth medium. Before the recording of the wounded area at 0 h, 24 h, 48 h, and 72 h, cells were stained with Hoechst 33342 (Sigma Aldrich, St. Louis, MO, USA) and captured with Cytation™ Cell Imaging Multi-Mode Reader (Biotek, Winooski, VT, USA). The changes of wounded areas were determined also by cell count analysis using Gene 5 software (Biotek, Winooski, VT, USA).

### 4.11. Co-Cultures Staining

For co-cultivation purposes, HeLa and BJ-5TA cells were seeded into 6-well plates in 1:1 ratio (0.04 × 10^6^:0.04 × 10^6^). After 24 h, cells were treated with L1 (1 µM and 10 µM) and incubated for 72 h. Incubation was followed by cell staining with Hoechst 33342 (Sigma Aldrich, St. Louis, MO, USA), N-cadherin, and Smad2/3 primary antibodies (all from Cell Signalling Technology^®^, Danvers, MA, USA). In the case of N-cadherin, goat anti-rabbit IgG secondary antibody conjugated with Alexa Fluor 488 (Thermo Scientific, Rockford, IL, USA) was used. For Smad2/3, cells were stained with goat anti-rabbit IgG secondary antibody conjugated with Alexa fluor 546 (Thermo Scientific, Rockford, IL, USA). The fluorescence signal was detected using automated Cytation™ 3 Cell Imaging Multi-Mode Reader (Biotek).

### 4.12. Statistical Analysis

Results are expressed as mean ± SD. Statistical analyses of the data were performed using standard procedures, with one-way ANOVA followed by the Bonferroni multiple comparisons test. Differences were considered significant when *p* < 0.05. Throughout this paper * indicates *p* < 0.05, ** *p* < 0.01, *** *p* < 0.001 vs. untreated control.

## 5. Conclusions

In conclusion, our study revealed the antiproliferative and pro-apoptotic effect of newly synthesized chalcone L1 against cervical cancer model. These results were followed by the involvement of L1 in the processes of EMT, including migration and invasiveness. According to the results, these effects were mediated by interfering with proteins involved in EMT (E-cadherin, N-cadherin, and MMP-9) or Smad-dependent and Smad-independent pathways, comprising Smad2/3 and MAP kinases. Moreover, L1 was able to modulate the effect of TGF-β and CM treatment favouring the antiproliferative effect, apoptosis and suppression of EMT in HeLa cells. Overall, results suggest the role of chalcones as a potential anti-cancer agents. However, further studies should be performed to determine underlying mechanisms involved in these processes.

## Figures and Tables

**Figure 1 molecules-26-01356-f001:**
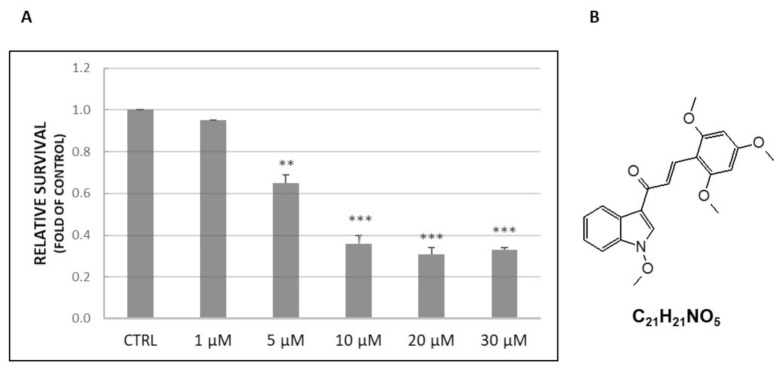
MTS assay (**A**) performed on HeLa cells treated with chalcone L1 (**B**). Representative data of three independent experiments are presented. Significantly different ** *p* < 0.01, *** *p* < 0.001 vs. untreated cells (control).

**Figure 2 molecules-26-01356-f002:**
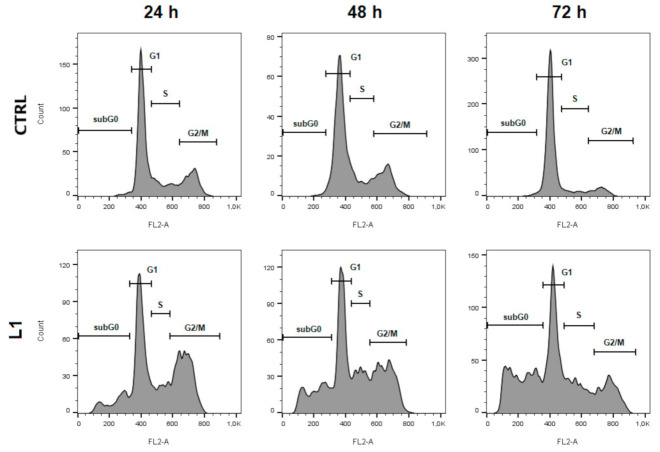
Histogram representation of cell cycle distribution in HeLa cells treated with L1 (10 µM) after 24 h, 48 h, and 72 h. Representative data of three independent experiments are presented.

**Figure 3 molecules-26-01356-f003:**
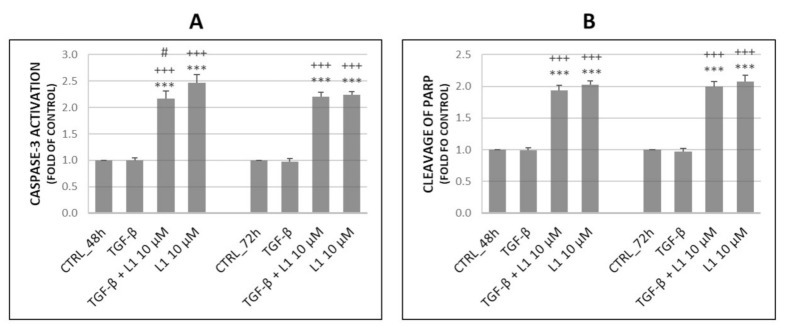
Changes in activation of Caspase-3 (**A**) and cleavage of PARP (**B**) obtained by flow cytometry on HeLa cells after 48 h and 72 h of L1 and TGF-β treatment. Representative data of three independent experiments are presented. Significantly different *** *p* < 0.001 vs. untreated cells (control); +++ *p* < 0.001 vs. TGF-β; # *p* < 0.001 vs. L1.

**Figure 4 molecules-26-01356-f004:**
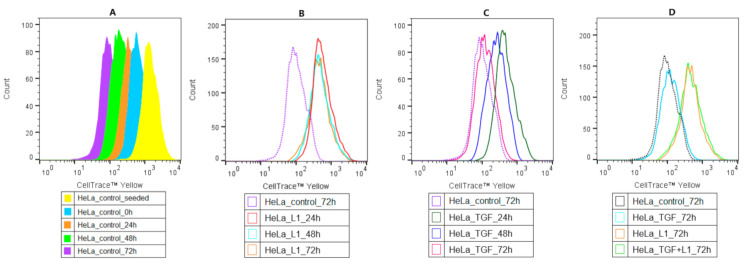
Flow cytometry analysis of HeLa cell proliferation stained with CellTrace™ Yellow. Control groups proliferation histograms represented by the figure (**A**), cells treated with L1 (10 µM) (**B**), cells treated with TGF-β (10 ng/mL) (**C**), cells treated with L1 or in combination L1/ TGF-β on 72 h (**D**). Representative histograms of three independent experiments are presented.

**Figure 5 molecules-26-01356-f005:**
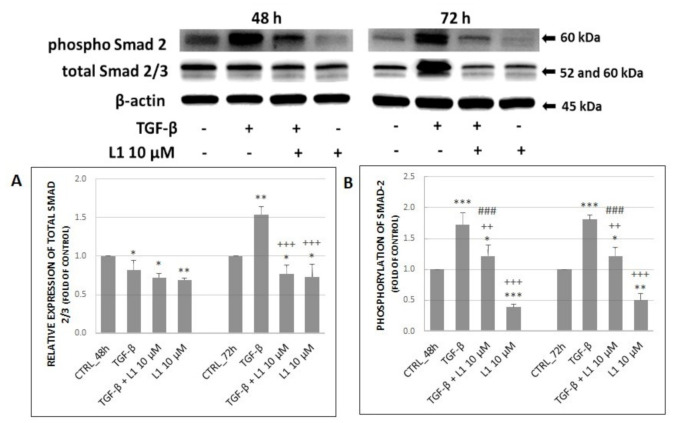
Changes in expression and phosphorylation of Smad proteins (**A**) along with corresponding densitometry analyses (**B**) of western blot results. All after 48 h and 72 h of L1 treatment. Representative data of three independent experiments are presented. Significantly different * *p* < 0.05, ** *p* < 0.01, *** *p* < 0.001 vs. untreated cells (control); ^++^
*p* < 0.01, ^+++^
*p* < 0.001 vs. TGF-β; ^###^
*p* < 0.001 vs. L1.

**Figure 6 molecules-26-01356-f006:**
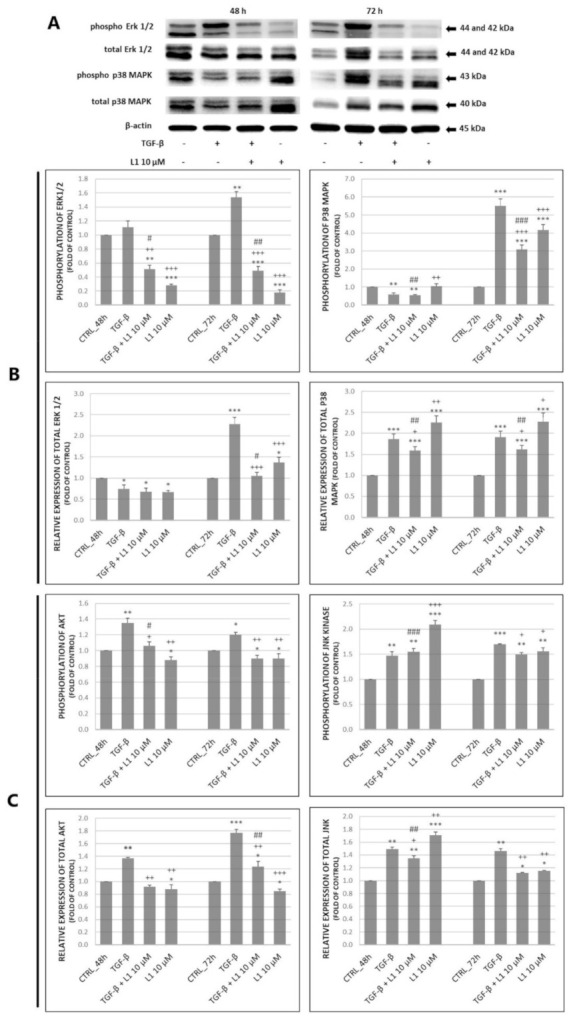
Changes in expression of MAPK related proteins (**A**) along with corresponding densitometry analyses (**B**) of western blot results. Relative level of Akt phosphorylation obtained by flow cytometry (**C**). All after 48 h and 72 h of treatment. Representative data of three independent experiments are presented. Significantly different * *p* < 0.05, ** *p* < 0.01, *** *p* < 0.001 vs. untreated cells (control); ^+^
*p* < 0.05, ^++^
*p* < 0.01, ^+++^
*p* < 0.001 vs. TGF-β; ^#^
*p* < 0.05, ^##^
*p* < 0.01, ^###^
*p* < 0.001 vs. L1.

**Figure 7 molecules-26-01356-f007:**
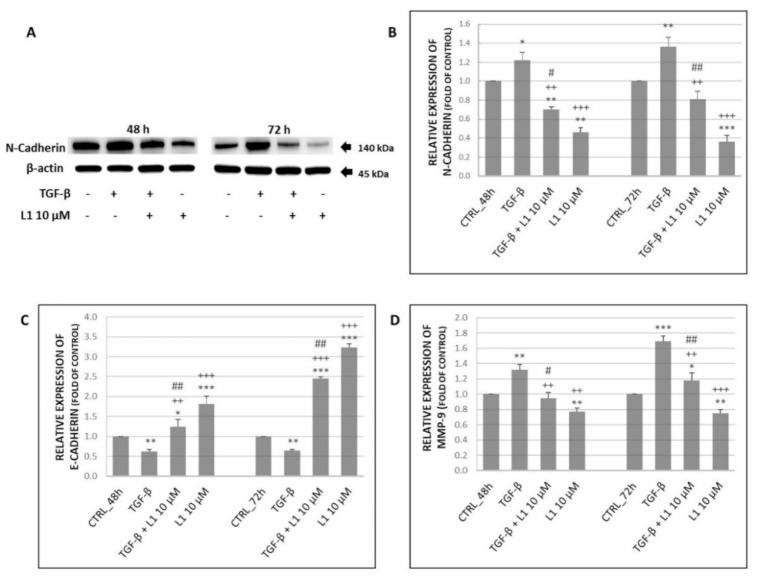
Changes in expression of N-Cadherin (**A**) along with its densitometry analyses (**B**) of western blot results. Relative expression of E-cadherin (**C**) and MMP-9 (**D**) obtained by flow cytometry analyses. All after 48 h and 72 h of L1 treatment. Representative data of three independent experiments are presented. Significantly different * *p* < 0.05, ** *p* < 0.01, *** *p* < 0.001 vs. untreated cells (control); ^++^
*p* < 0.01, ^+++^
*p* < 0.001 vs. TGF-β; ^#^
*p* < 0.05, ^##^
*p* < 0.01 vs. L1.

**Figure 8 molecules-26-01356-f008:**
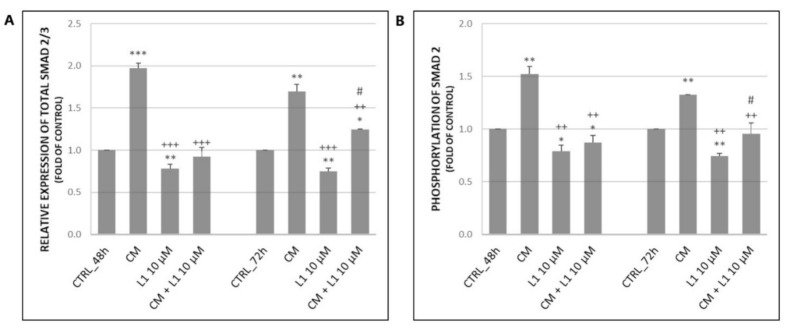
Changes in expression (**A**) and phosphorylation (**B**) of Smad proteins after 48 h and 72 h of CM from BJ-5ta cells and L1 treatment. Representative data obtained by three independent experiments using flow cytometry. CM—conditioned medium. Significantly different * *p* < 0.05, ** *p* < 0.01, *** *p* < 0.001 vs. untreated cells (control); ^++^
*p* < 0.01, ^+++^
*p* < 0.001 vs. CM; ^#^
*p* < 0.05 vs. L1.

**Figure 9 molecules-26-01356-f009:**
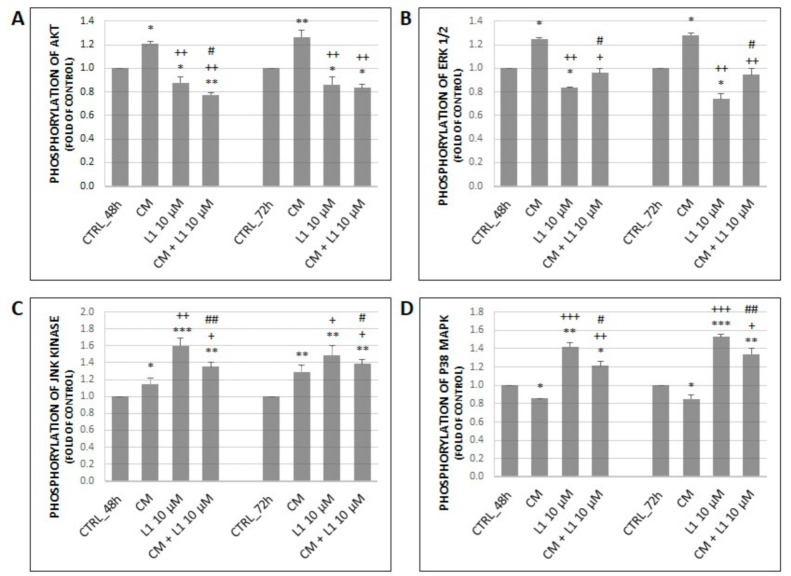
Changes in phosphorylation of Akt (**A**) and MAPK related proteins including Erk1/2 (**B**), JNK (**C**), and p38 (**D**) after 48 h and 72 h of CM from BJ-5ta cells and L1 treatment. Representative data obtained by three independent experiments using flow cytometry. CM—onditioned medium. Significantly different * *p* < 0.05, ** *p* < 0.01, *** *p* < 0.001 vs. untreated cells (control); ^+^
*p* < 0.05; ^++^
*p* < 0.01, ^+++^
*p* < 0.001 vs. CM; ^#^
*p* < 0.05; ^##^
*p* < 0.01 vs. L1.

**Figure 10 molecules-26-01356-f010:**
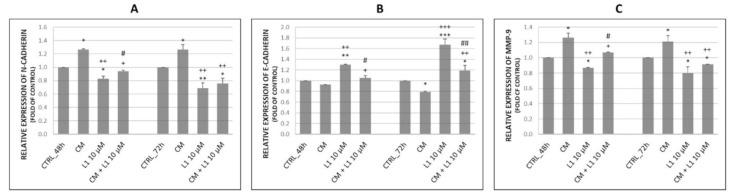
Changes in expression of N-Cadherin (**A**), E-Cadherin (**B**), and MMP-9 (**C**) after 48 h and 72 h of CM from BJ-5ta cells and L1 treatment. Representative data obtained by three independent experiments using flow cytometry. CM—conditioned medium. Significantly different * *p* < 0.05, ** *p* < 0.01, *** *p* < 0.001 vs. untreated cells (control); ^+^
*p* < 0.05; ^++^
*p* < 0.01, ^+++^
*p* < 0.001 vs. CM; ^#^
*p* < 0.05; ^##^
*p* < 0.01 vs. L1.

**Figure 11 molecules-26-01356-f011:**
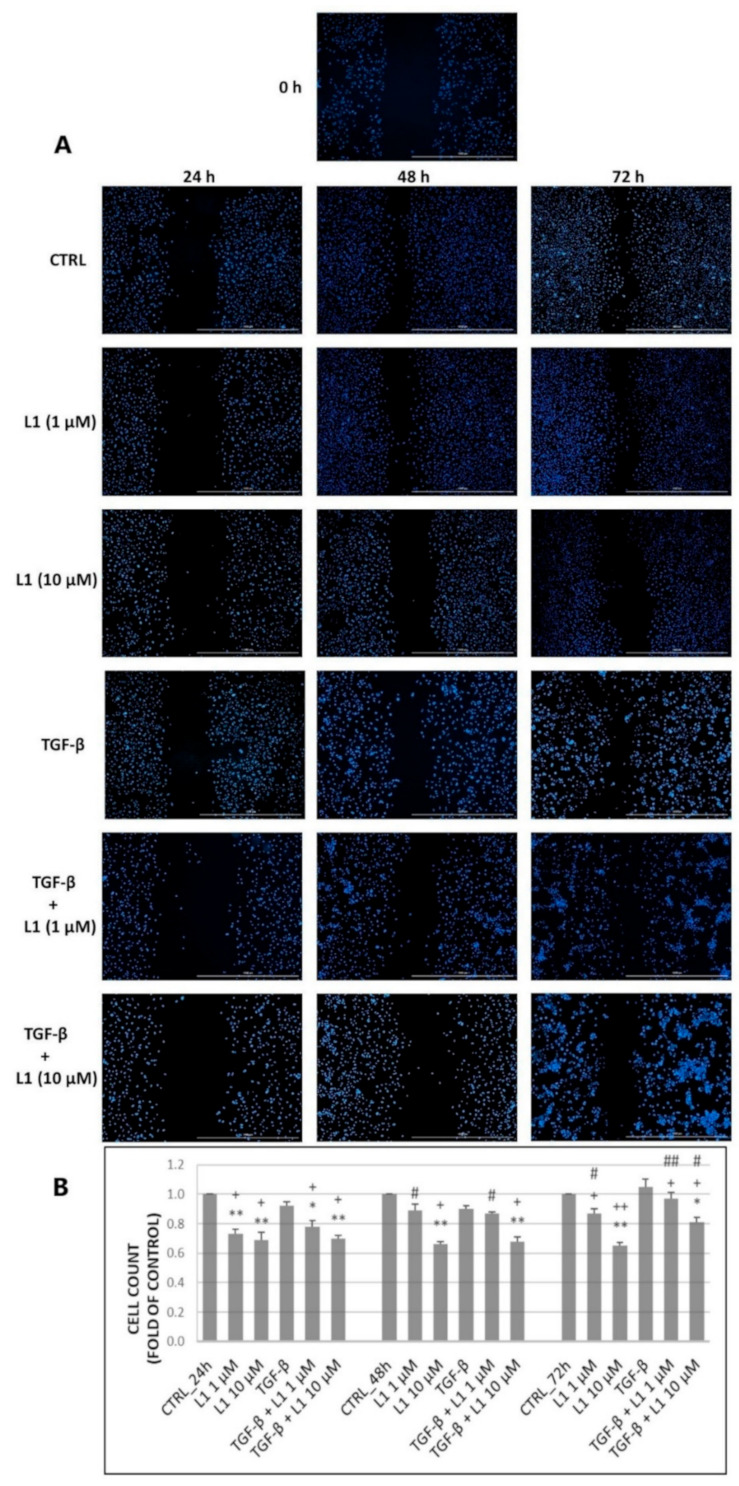
Scratch assay performed on HeLa cells treated with 1 or 10 µM concentration of L1 and TGF-β (10 ng/mL) or their mutual combinations compared to control groups at three different times (24 h, 48 h, and 72 h) (**A**). Graph representation of cell count in individual figures of part A (**B**). Representative data obtained by three independent experiments. CTRL—control groups. Significantly different * *p* < 0.05, ** *p* < 0.01 vs. untreated cells (control); ^+^
*p* < 0.05, ^++^
*p* < 0.01 vs. TGF-β; ^#^
*p* < 0.05, ^##^
*p* < 0.01 vs. L1 (10 µM).

**Figure 12 molecules-26-01356-f012:**
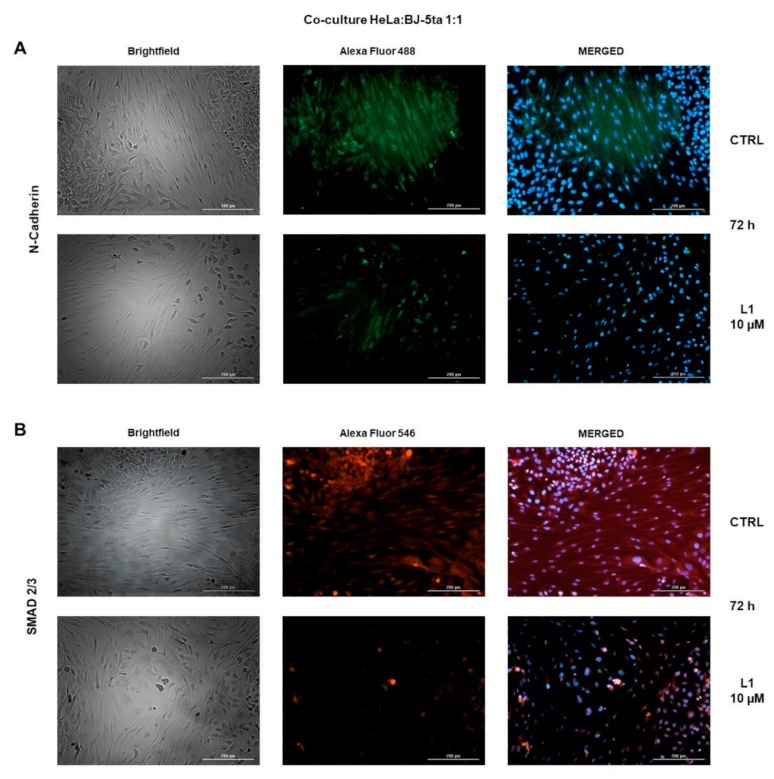
Representative HeLa and BJ-5ta co-culture staining pictures of N-Cadherin (**A**) and Smad2/3 (**B**) proteins at 72 h captured by live cell imaging.

**Table 1 molecules-26-01356-t001:** Cell Cycle Analysis of HeLa Cells after 24 h, 48 h, and 72 h Incubation with L1.

	Sub-G0	G1	S	G2/M
**CTRL 24 h**	2.19 ± 0.06	61.25 ± 2.65	17.25 ± 0.29	19.30 ± 2.29
**L1**	8.04 ± 2.08	53.70 ± 0,80 *	13.80 ± 0.49	24.45 ± 1.78 *
**CTRL 48 h**	1.77 ± 0.40	69.35 ± 3.13	12.90 ± 1.40	15.95 ± 1.28
**L1**	11.36 ± 0.65 *	54.45 ± 3.43 *	13.41 ± 2.85	20.8 ± 0.45 *
**CTRL 72 h**	2.17 ± 0.31	78.35 ± 0.29	9.25 ± 0.29	10.21 ± 0.32
**L1**	18.05 ± 0.40 **	55.05 ± 2.85 **	13.27 ± 3.05	13.65 ± 2.25

The data are presented from three independent experiments after 24 h, 48 h, and 72 h of L1 treatment as mean percentage ± SD. Significantly different * *p* < 0.05, ** *p* < 0.01 vs. untreated cells (control).

**Table 2 molecules-26-01356-t002:** Flow Cytometric Analysis of Annexin V/PI Staining Performed on HeLa Cells after 24, 48 and 72 h of L1 and TGF-β Treatment.

	Live	Early Apoptotic	Late Apoptotic	Death
**CTRL 24 h**	92.95 ± 0.55	2.05 ± 0.41	1.55 ± 0.05	3.46 ± 0.94
**TGF-β**	94.00 ± 1.10	1.70 ± 0.70	1.70 ± 0.26	2.87 ± 0.16
**TGF-β + L1**	88.40 ± 3.40	4.04 ± 2.62	3.90 ± 0.61	3.66 ± 0,18
**L1**	85.00 ± 0.40 *+	6.02 ± 0.60	4.94 ± 0.73	4.03 ± 0.26
**CTRL 48 h**	93.30 ± 0.50	2.44 ± 0.02	3.04 ± 0.56	1.25 ± 0.10
**TGF-β**	95.05 ± 0.35	2.28 ± 0.20	1.61 ± 0.49	1.03 ± 0.07
**TGF-β + L1**	83.20 ± 1.60 *+	11.53 ± 2.73 *+	3.52 ± 0.62	1.76 ± 0.13
**L1**	81.70 ± 0.70 *+	11.89 ± 0.72 *+	4.42 ± 1.15	1.95 ± 0.27
**CTRL 72 h**	93.20 ± 0.70	2.19 ± 0.18	3.09 ± 0.67	1.52 ± 0.26
**TGF-β**	92.95 ± 1.65	3.24 ± 1.51	2.36 ± 0.13	1.48 ± 0.26
**TGF-β + L1**	76.65 ± 1.35 **++	15.91 ± 0.41 *+	6.64 ± 0.41	4.99 ± 0.53
**L1**	74.35 ± 1.45 **++	17.34 ± 1.54 **++	2.43 ± 0.11	1.66 ± 0.20

The data are presented from three independent experiments after 24 h, 48 h, and 72 h of L1 (10 µM) and TGF-β treatment as mean percentage ± SD. Significantly different * *p* < 0.05, ** *p* < 0.01 vs. untreated cells (control); ^+^
*p* < 0.05, ^++^
*p* < 0.01 vs. TGF-β.

**Table 3 molecules-26-01356-t003:** Antibodies Used for Flow Cytometry Staining.

**Primary Antibodies**	
Phospho-Smad2 (PE Conjugate)	Cell Signalling Technology^®^ Danvers, MA, USA
Smad2/3 (PE Conjugate)	
Phospho-Akt (PE Conjugate)	
Akt Rabbit mAb	
Phospho-p44/42 MAPK (Erk1/2) (Alexa(R)488)	
Phospho-p38 MAPK (PE Conjugate)	
Phospho-SAPK/JNK Rabbit mAb	
E-Cadherin (Alexa(R)488)	
N-Cadherin (Alexa(R)647)	
MMP-9 (PE Conjugate)	
Caspase-3 (PE Conjugate)	
PARP (PE Conjugate)	
Annexin V (Alexa(R)546)	Thermo Scientific Rockford, IL, USA
JNK 1 Mouse mAb	
**Secondary Conjugated Antibody**	
Alexa Fluor^TM^ 488 goat anti-rabbit IgG	Thermo Scientific Rockford, IL, USA
Alexa Fluor^TM^ 488 goat anti-mouse IgG	

**Table 4 molecules-26-01356-t004:** Antibodies Used for Immunoblotting.

**Primary Antibody**	**Mr (kDa)**	**Source/Origin**	**Company**
β-actin	45	Mouse	Cell Signalling Technology^®^
Phospho Smad2	60	Rabbit	Thermo Scientific
Smad2/3	52/60	Rabbit	Cell Signalling Technology^®^
N-Cadherin	140	Rabbit	
Phospho p44-42 MAPK (Erk1/2)	44/42	Mouse	
p44-42 MAPK (Erk1/2)	44/42	Rabbit	
Phospho p38 MAPK	43	Rabbit	
p38 MAPK	40	Rabbit	
**Secondary Antibody**	**Mr (kDa)**	**Source/Origin**	**Company**
Anti-mouse IgG HRP	-	Goat	Cell SignallingTechnology^®^
Anti-rabbit IgG HRP	-	Goat	

## Data Availability

The data presented in this study are available on request from the corresponding author.

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
