# Peer review of "The Newly Synthetized Chalcone L1 Is Involved in the Cell Growth Inhibition, Induction of Apoptosis and Suppression of Epithelial-to-Mesenchymal Transition of HeLa Cells"

_molecules, 2021, doi:10.3390/molecules26051356_

Round 1

Reviewer 1 Report

The authors demonstrated that newly synthesized chalcone L1 exhibited antiproliferative effect, induced apoptosis, and suppressed EMT in HeLa cells through Smad-dependent and Smad-independent pathways. The diverse approach to verify your results is impressive. However, more evidence is needed to support your conclusion. The parts that should be revised are shown below.

  1. The results of cell cycle analysis in Figure 2 are not sufficient to prove apoptosis induction. Please present another evidence, such as Tunnel assay, annexin V/PI double staining assay, and western blot (cleaved PARP, cleaved caspase-3).
  2. In Figure 3, it seems that the proliferation of TGF-treated group was not increased compared with that of untreated group (72 h). This is not consistent with your description in line 135-138.
  3. In Figure 4-9, the authors sometimes conducted western blot and sometimes performed flow cytometry to evaluate the expression of the indicated proteins without any explanation of the reason. For example, in Figure 5, the expression of p-p38, t-p38, p-ERK and t-ERK was measured by western blot, whereas that of p-AKT was detected by flow cytometry. Why? It would be desirable if the authors can present western blot results for all experiments.
  4. In Figure 5, why are there no results of t-AKT, p-JNK, and t-JNK?
  5. In Figure 6 legend, there is no description about Figure 6D.
  6. In the descriptions of Figure 7-9, please specify the name of fibroblast cell line.
  7. In Figure 10, why didn’t you treat TGF-beta or conditioned media to stimulate the migration of cancer cells as the other experiments? I think you should add the results. In addition, as shown in Figure 1, 10 μM of chalcone L1 decreased the cell viability of HeLa cells. Therefore, the decreased migration by chalcone L1 could be due to the cytotoxicity.

Author Response

Dear reviewer, we thnak you for your valuable comments. See please answers in attached file.

Best regards

Kello

Reviewer 2 Report

The author clearly explored the anti-tumor and anti-metastatic mechanism of newly synthetized chalcone L1. However, there are some questions that need to be confirmed.

  1. Line 104, the author showed “ IC50 value is represented by a cca 10 uM chalcone L1”. Please explain what is cca?
  2. Line 301, there seems to be a paragraph to explain the effect of L1 on selected protein, but only one sentence description.
  3. Figure 6A seems to have adjusted the contrast or background modification. This result has a significant difference in different treatments. I suggest this result does not require background modification.
  4. Image results in Figure 10 and Figure 11 are unclear. Please provide high-quality pictures.

Author Response

Dear reviewer,

we thank you for your valuable comments and suggestion. For our response please see attached file.

Round 2

Reviewer 1 Report

I think the manuscript has been well-revised according to the reviewer's comments.

Reviewer 2 Report

All concerns have been addressed.